# Analysis of Attitudes and Practices towards the Influenza Vaccine in High-Risk Adults in Poland

**DOI:** 10.3390/vaccines12030341

**Published:** 2024-03-21

**Authors:** Dorota Kopciuch, Karolina Hoffmann, Krzysztof Kus, Agnieszka Koligat-Seitz, Piotr Ratajczak, Elżbieta Nowakowska, Anna Paczkowska

**Affiliations:** 1Department of Pharmacoeconomics and Social Pharmacy, Poznan University of Medical Sciences, Rokietnicka 3 Street, 60-806 Poznan, Poland; kkus@ump.edu.pl (K.K.); p_ratajczak@ump.edu.pl (P.R.); aniapaczkowska@ump.edu.pl (A.P.); 2Department of Internal Diseases, Metabolic Disorders and Arterial Hypertension, Poznan University of Medical Sciences, Szamarzewskiego 84 Street, 60-572 Poznan, Poland; karolinahoffmann@ump.edu.pl; 3Department of Otolaryngology, Greater Poland Pediatric Center, Wrzoska 1 Street, 60-663 Poznan, Poland; aga.kol@vp.pl; 4Department of Pharmacology and Toxicology, Institute of Health Sciences, Collegium Medicum, University of Zielona Gora, Licealna 9 Street, 65-417 Zielona Góra, Poland; elapharm@ump.edu.pl

**Keywords:** vaccines, influenza, high-risk groups, attitudes, practice

## Abstract

This study aimed to understand Poles’ attitudes and beliefs towards influenza vaccinations in the flu season of 2022–2023, especially among individuals at risk of flu complications. The cross-sectional survey-based study was carried out on a sample of 810 respondents. The questionnaire was disseminated electronically using social media and e-mail. The majority of respondents (71%) could identify “high-risk groups” recommended for influenza vaccination, and 52.01% of respondents reported receiving influenza vaccination at some point in the past, with 32.12% receiving it in the 2022–2023 flu season and 41.09% in the 2021–2022 season. The majority of respondents declaring acceptance of the vaccine for the 2022–2023 season were in the high-risk group. Only 17.28% of respondents declared receiving both influenza and SARS-CoV-2 vaccines in the 2022–2023 season, with the vast majority being respondents from the “high-risk group” (*p* < 0.0001). Only 26.12% of respondents declared their intention to continue influenza vaccination in the future. Of those expressing the intention to get vaccinated against the influenza virus in future seasons, 46.79% were from the “high-risk group” (*p* = 0.0087). Results suggest the need for further interaction and education with healthcare providers and targeted informational efforts for at-risk groups regarding the benefits of flu vaccination.

## 1. Introduction

Annual influenza epidemics are a serious public health problem, causing particularly severe morbidity and mortality in high-risk groups. Children are among the high-risk groups, especially those under 2 years of age, as well as persons over 65 and patients with underlying medical conditions that may make them more susceptible to complications from the flu [1,2]. The World Health Organization (WHO) and European Centre for Disease Prevention and Control (ECDC) recommend annual influenza vaccinations for selected groups of patients, especially for pregnant women, children aged between 6 months and 5 years, and older patients, specifically those above 65 years of age. In the patient group with a high risk of flu-related complications, individuals with chronic diseases, as well as healthcare workers, are also included [3,4]. The recommendations are analogous to those of the scientific community [5,6,7]. Regardless of recommendations, influenza vaccination coverage varies depending on the country [3,4]. Studies indicate that the level of flu cases and influenza coverage in the European regions of the WHO during the 2021–2022 season was relatively high and also varied by country [8]. Currently, influenza vaccination rates are below optimal levels, with only a few developed countries reaching the WHO/European Council target of 75% vaccination coverage in persons aged 65 years and above [9]. The vaccination coverage rate in 19 member states in 2016–2017 showed that the vaccination coverage rates in older target populations ranged from 2.0% to 72.8% (median 47.1%) [10]. Reports from Poland suggest that the attitude of Poles towards influenza vaccinations does not align with the actual vaccination coverage, as despite a positive opinion regarding the effectiveness and importance of vaccinations, the percentage of vaccinated individuals remains low [10,11,12,13]. 

Poland is a country with one of the lowest influenza vaccination coverage rates among the EU member states [4]. In Poland, influenza vaccination is voluntary. Public health authorities largely align with those guidelines specified by the WHO and ECDC. Children aged over 6 months to under 18 years old, individuals aged 65–74, and individuals aged 18–65 with chronic illnesses are eligible to purchase the vaccine with a 50% reimbursement. Pregnant women and individuals aged 75 and above are entitled to obtain the influenza vaccine free of charge [14]. Furthermore, for free influenza vaccines in the 2021/2022 season, the following were authorized: 

-Individuals employed in healthcare entities, pharmacies, and diagnostic laboratories;-All academic teachers, as well as students and doctoral candidates participating in activities involving patients;-Individuals employed in the State Pharmaceutical Inspection Authority, regional inspectors, and the Chief Pharmaceutical Inspector;-Individuals employed in social welfare units or facilities providing round-the-clock care for disabled persons, chronically ill persons, or elderly persons;-Patients of care-and-treatment facilities, nursing-care facilities, hospices, and palliative care units;-Individuals residing in care homes, facilities for disabled persons, chronically ill persons, or elderly persons;-Individuals working in kindergartens, schools, or other educational institutions, care–educational and care–therapeutic institutions, and pre-adoption centers;-Uniformed service officers and soldiers;-Patient rights advocates of psychiatric hospitals;-Individuals born no later than 1946 [15].

Despite clear recommendations and the high availability of influenza vaccinations, the vaccination rate has remained consistently low for many years. For example, during the influenza season of 2021–2022, the Government Strategic Reserves Agency (an organization responsible for supplying the public market) distributed purchased vaccines to vaccination points for the free provision of services to all adults. Influenza vaccines were available to all adult Poles even without the need for a prescription (normally required). Since 11 January 2022, pharmacists can administer influenza vaccines in pharmacies, but they can only issue prescriptions pro auctore and pro familiae [16]. Despite being reimbursed, influenza vaccinations do not have a fixed reimbursement amount; the level of co-payment varies. Vaccines covered by reimbursement must be purchased at a pharmacy because those offered at Healthcare Facilities are fully paid products. This is due to the lack of Healthcare Facilities’ authorization to trade in medicinal products, especially those eligible for reimbursement. This is a significant limitation in utilizing Healthcare Facilities’, as they were, until recently, the default locations for vaccinations. Currently, pharmacies are increasingly chosen as the preferred locations for vaccinations, potentially becoming the main sites for publicly funded vaccinations in the future. However, a significant limitation remains the prohibition of advertising services provided by a pharmacy [16,17]. 

According to data published by the Polish Chief Sanitary Inspectorate, the percentage of people vaccinated against the flu during the 2019/2020 flu season was 4.12%, while in the previous season (2018/2019) it was 3.9%. On the other hand, influenza vaccination coverage among people aged 65+ increased over the decade from 12.5% to 22.6%. The low level of vaccination coverage thus indicates a low level of public awareness of the dangers of not being vaccinated. This may also mean that the public does not understand the benefits of vaccination, which, in turn, may result from a lack of health education and health promotion [18]. The situation is even more serious as we know that the world is still struggling with the current SARS-CoV-2 virus, which may result in additional health burdens [19]. The awareness of the role of vaccinations in ensuring individual and population health safety plays a significant role in this regard. 

This study put forward the following hypotheses: 1. Despite a high level of knowledge about the role of flu vaccinations, the vaccination coverage rate remains very low. 2. Patients from the high-risk group undergo flu vaccinations much more frequently compared to patients outside this group. 3. The main reason for not getting vaccinated against the flu is a lack of trust in the safety of vaccines

This study aimed to understand Poles’ attitudes and practices towards influenza vaccinations in the flu season of 2022–2023, analyze preferences and unmet needs regarding influenza vaccinations, and examine the perception and acceptance of the influenza vaccine among individuals at risk of flu complications. The results of this study may be helpful in the development of future educational programs that will contribute to increasing public awareness and thus increasing the percentage of people vaccinated against the influenza virus in the coming years.

## 2. Materials and Methods

A prospective, cross-sectional web-based survey design was adopted. The study horizon was from 1 July 2022 to 31 July 2023. The survey was performed using the Computer-Assisted Web Interview (CAWI) method. The Google Forms survey was created by the study authors and distributed electronically. It was a fully voluntary study that used an anonymous questionnaire for all individuals aged 18 or older who live in Poland and have access to the Internet. The questionnaire in Polish was distributed mainly via Facebook and e-mail. This study utilized a database held by the academic institution of the researchers, consisting of email addresses (2160 records). One thousand emails were sent, and the link was shared on Facebook 1000 times. The messages were sent to the respondents with a request to join the study voluntarily. The messages contained a link to the research questionnaire and all necessary information about the study’s purpose and rules of participation.

A literature review was conducted before designing the questionnaire. Important questions and topics from the literature were either modified or directly included as items in our questionnaire. A pre-test procedure on a representative sample of 200 subjects was used [20]. After the procedure, if necessary, the questions could be modified. However, the pre-test results were included in the post-test because the pre-test showed no need to modify the study tool.

The questionnaire (Appendix A) comprised 25 questions divided into 4 categories: demographic data (age, sex, education, place of living); medical anamneses (chronic illnesses, health status); COVID-19-related and influenza anamneses (number of vaccine doses, previous infection, side effects); and correlation between vaccination against influenza and COVID-19 and attitude on general influenza vaccination. Most of the questions involved multiple variables, and the number of questions visible to respondents depended on their answers, as some (but not all) responses led to additional questions.

The survey response rate was 40.5%. Out of 1446 completed questionnaires, 810 respondents were included for further analysis. A total of 636 respondents were excluded from further analysis based on exclusion criteria (Figure 1). A prespecified high-risk subgroup included respondents who met at least one of the following criteria: ≥65 years of age, pregnancy, residing in a long-term care facility, Indigenous origin, or a high-risk condition as defined by the Centers for Disease Control and Prevention (CDCP) (i.e., hypertension, diabetes, or other metabolic diseases, asthma, other chronic lung diseases, heart diseases, body mass index [BMI] above 40 kg/m^2^, anemia, immune disorders, cancer, or other conditions negatively affecting respiratory function and/or increasing the risk of aspiration). 

Statistical analysis was performed using STATISTICA PL 10.0 (StatSoft).

The data distribution pattern was not normal (unlike the Gaussian function)—calculations were performed using the chi-square test. The analysis of the Test for Proportions determined significant differences between % of group results for *p* < 0.05. The level of statistical significance was set at *p* < 0.0083 (using the Bonferroni correction) to avoid an excessive number of Type I errors.

## 3. Results

### 3.1. Study Population

This study involved 1446 respondents. After considering the previously discussed inclusion criteria, responses from 810 participants were included for further analysis (Figure 1). The study group comprised 467 women and 343 men aged 18 to 87 years. The average age was estimated at 38.32 years (Table 1). Individuals defined as belonging to the “high-risk group” with a medical background constituted 38.23% of the study population (Table 1). The entire high-risk cohort accounted for 58.23% of the study population, including individuals aged 65 and older, pregnant women, and those with a high medical risk. 

### 3.2. Knowledge

The majority of respondents (71%) could identify “high-risk groups” recommended for influenza vaccination, with a significant majority being respondents defined as patients in the “high-risk group” (77.83%) (*p* < 0.0001) (Table 2). Furthermore, it was observed that individuals aged 21 to 59 demonstrated a higher level of knowledge in this regard compared to those below 20 and above 65 years old (Table 2). Additionally, education level was a differentiating factor in knowledge about risk groups, as individuals with higher and medium education levels were much better at identifying risk groups for influenza virus infection. It was also statistically significant, and the strength of significance varied depending on the represented age group (Table 2).

### 3.3. Attitudes and Practices

Only 36% of respondents declared being diagnosed with influenza virus infection in the 2022–2023 season, with a significant majority being respondents from the “high-risk group” (*p* < 0.0001), individuals above 65 years old (*p* < 0.05), and those with higher education (*p* ≤ 0.001) (Table 2).

The majority of respondents (52.01%) reported receiving influenza vaccination at some point in the past, with 32.12% receiving it in the 2022–2023 flu season and 41.09% in the 2021–2022 season. 

The majority of respondents declaring acceptance of the vaccine for the 2022–2023 season were in the high-risk group, and it was also observed that individuals above 65 years of age were more likely to undergo influenza vaccination. Interestingly, no statistically significant differences were observed in the observed results based on the respondent’s level of education (Table 2).

Only 28.08% of them experienced adverse events (AEs), with 65.28% having mild AEs, 31.22% having moderate AEs, and one patient experiencing severe AEs (Table 2).

Only 17.28% of respondents declared receiving both influenza and SARS-CoV-2 vaccines in the 2022–2023 season, with the vast majority being respondents from the “high-risk group” (*p* < 0.0001) (Table 2). Most of them received three doses of the SARS-CoV-2 vaccine. These observations were not statistically significant. Furthermore, 67.09% of respondents believed that the flu vaccine they received alleviates the symptoms of a SARS-CoV-2 infection, with a significant majority of those agreeing to be from the “high-risk group” (*p* = 0.001), and these observations were age dependent, coming from older individuals (Table 2).

### 3.4. Reasons for Getting/Not Getting Vaccinated

Only 26.12% of respondents declared their intention to continue influenza vaccination in the future, while 27.88% were uncertain about taking such action (Table 2). Of those expressing the intention to get vaccinated against the influenza virus in future seasons, 46.79% were from the “high-risk group” (*p* = 0.0087) (Table 2). 

The main reason for respondents opting for influenza vaccination was the fact that they or their close ones belonged to the “high-risk group.” Additionally, individuals in the “high-risk group” declared that getting vaccinated is a matter of social responsibility and that it significantly influences their decision, as vaccinations alleviate the symptoms of other diseases, including viral ones (Figure 2). On the other hand, the primary reason for individuals outside the “high-risk group” to undergo vaccinations was the frequent occurrence of influenza in themselves or their close ones (Figure 2).

Among those expressing unwillingness to get vaccinated against influenza in future seasons, the majority were respondents outside the “high-risk group” (*p* = 0.0057). These observations were also correlated with education level, with those with lower education levels more frequently reporting a lack of willingness to undergo vaccinations in the future (*p* < 0.05) (Table 2).

The main reasons for respondents not wanting to undergo influenza vaccination were fear of adverse events and doubts about the effectiveness of vaccinations (Figure 3).

## 4. Discussion

In the WHO European Region, annual influenza epidemics typically occur in the fall and winter [21]. 

In Poland, to reduce financial barriers, refunds for influenza vaccine costs were introduced in 2018, including full reimbursement for selected groups from 2020 [22]. 

However, despite increased access to vaccinations, during the 2020/2021 flu season, only a negligible percentage of Poles were vaccinated against seasonal influenza [23]. Interestingly, more Poles (13.4%) declared their willingness to be vaccinated against the flu in the 2020/2021 season than the actual number of those vaccinated (6%). Such observations serve as evidence of existing disparities between the attitudes of Poles towards vaccinations and the practices they undertake. 

We can also speculate that such a significant difference between the percentage of respondents declaring their intention to be vaccinated against the flu and the actual number of those vaccinated may not only stem from educational barriers but also financial or systemic ones. In addition to ensuring access to vaccines, the opportunity for vaccination near one’s home and the elimination of financial barriers and health policy programs should focus on educating about the “patient path” and removing all kinds of restrictions. Studies indicate that the highest vaccination rates are observed among individuals aged 65 and above. However, during the epidemic season of 2019/2020 in Poland, vaccine uptake in this group was only 10.4%, significantly below the average coverage in EU member states (44%). This aligned with the results of our study, as only 18.45% of respondents declared ever receiving the vaccine. On the other hand, vaccine acceptance for the 2022–2023 season was declared by 32.12%. Similar observations were made by the authors of other studies [24], which indicated that 39% of all adults, 27% of adults aged 18–64 without chronic diseases, 38% in the age group of 18–64 with chronic diseases, and 71% of adults aged 65 and above were vaccinated against the flu.

The findings of our study also suggest certain relationships, such as the presented level of knowledge, attitudes towards vaccination, and vaccination practices. In our study, a relatively high level of knowledge was observed regarding the identification of factors and risk groups for influenza, translating into a proactive approach to vaccination. Respondents who could identify risk groups tended to express a greater willingness to undergo vaccination. This was particularly evident among older individuals who belong to a high-risk group. The reports contrast with observations from other authors [25,26,27], where it was assessed that although the majority of adults under the age of 65 at high risk were familiar with the recommended guidelines for flu vaccination, only 35.8% (95% CI, 32.1–39.5%) were aware of their high-risk status. In our study, it was additionally observed that qualification for the “high-risk group” influences respondents’ attitudes towards getting protective vaccinations, as individuals above 65 years old in the high-risk group had a more positive attitude towards getting flu vaccinations compared to respondents in the same age range who were not at risk. Furthermore, 33% of respondents in the high-risk group and 30% outside this group believed that flu vaccination is important for individuals at risk. Similar proportions are observed when analyzing the motivations for respondents not getting vaccinated against the flu, with 36% and 35% expressing concerns about adverse events (Aes), respectively, among patients at risk and those not at risk. This forms the basis for considering the development of awareness programs for society regarding the safety profile of available vaccines, as well as the significant advantages of vaccination [28]. One of the most commonly reported mild adverse events was fatigue, which was also observed by the authors of other studies [29,30].

The vaccination rates reported in this study can be considered an assessment of the knowledge and perceptions of respondents regarding influenza vaccination in Poland. This issue has also been addressed by authors in other studies [31,32,33,34]. The fact that only 32.12% of respondents received the flu vaccine in the current season, compared to 41.09% in the previous season, suggests that many Poles may not be aware of the benefits of flu vaccination. It is also concerning that as many as 23% of at-risk respondents did not get vaccinated against the flu due to trivial reasons, such as a lack of time. Considering the high availability of vaccines, these results suggest that education of the society plays a crucial role in improving vaccination coverage rates, especially if these efforts are targeted at high-risk individuals. Among those declaring vaccine acceptance, older individuals (over 65 years old) were significantly predominant, aligning with the vaccination coverage goals of 70% according to the WHO. Compared to adults under 65 years old (with or without chronic diseases), older individuals not only more frequently reported being vaccinated but also had a stronger belief in the importance of flu vaccination for high-risk groups. These results are corroborated by the authors of another study [31,32,35]. 

It is worth noting that although vaccination rates were relatively high among older individuals in our study, 34.12% of them did not know or were unsure if they were at risk of flu complications. Our study did not inquire about the types of vaccines respondents received in the 2022–2023 season, so it is unknown whether older participants in this study received the standard flu vaccine or an enhanced one specially designed for their age group. Overall, our results in both younger and older high-risk groups suggest the need for greater education and communication between healthcare decision makers and the community to increase awareness of the importance of flu vaccination, especially among those at risk of complications. For concurrent COVID-19 and flu vaccination in the 2022–2023 season, the vaccination rate was estimated at a very low level of 17.28%. This rate was significantly lower than in other studies [25].

Low co-administration rates were not associated with a lack of awareness, as most respondents were aware that both vaccines could be administered simultaneously. The majority of respondents also expressed interest in the concurrent administration of vaccines, especially those aged 65 and above. Such observations suggest an urgent need to develop educational programs to increase awareness among Poles about the benefits of receiving vaccines during the flu season and in the era of ongoing COVID-19 infections.

A significant challenge in managing influenza vaccinations in Poland is the lack of adequate and reliable information. As influenza experts in Poland point out, information about individuals vaccinated against the flu is limited. This is because not every influenza vaccination is recorded, and this only applies to those funded by public funds. This results in gaps in [16] knowledge about the number of vaccinations administered and systemic information about the immunization status of all patients. The vaccination system itself is decentralized and relies on Healthcare Facilities and, since 2022, also on pharmacies. Each of these institutions maintains its independent appointment scheduling system. In the case of influenza, there are no referrals for vaccinations. In many countries worldwide, referrals serve nurses, midwives, and pharmacists to assess patients’ immunization needs and administer various vaccinations, including those funded by public funds [36]. The previously described imperfect reporting system for influenza vaccinations largely limits planning capabilities. The pandemic has changed the attitudes of a significant portion of adult Poles towards influenza vaccinations, with declarative data indicating a significant increase in willingness to get vaccinated. Therefore, influenza vaccinations should be reported in the same manner as COVID-19 vaccinations. Many European countries have statistics on influenza vaccinations administered to various health and epidemiological risk groups [37]. It is important not to forget that knowledge about patients’ immunization status, beyond the organizational and planning sphere, can also have significant clinical and diagnostic value. Well-organized vaccinations from the political sector enable healthcare providers to plan the influenza season. Therefore, decisions regarding the rules and availability of vaccinations should be made and announced well before the vaccination season, allowing all stakeholders to prepare adequately. The next issue is the financial difficulties faced by Polish society. Rising living costs and limited financial resources may lead people to make difficult decisions regarding healthcare expenses, including vaccinations against the flu. Consumer studies can provide valuable data on the health priorities of individuals in Poland in the face of financial difficulties. These data may include an analysis of whether people are willing to invest in disease prevention through vaccinations or if they are focusing on more urgent financial needs. Depending on the availability of vaccinations, their financial accessibility, and societal awareness of the health benefits of flu vaccinations, individuals in Poland may make decisions regarding receiving these vaccinations. Health priorities in the face of financial difficulties may also influence decisions by the government and healthcare institutions regarding the promotion, access, and reimbursement of flu vaccinations.

Because this study was designed as a survey, the presented results are subject to several limitations. The main constraint of this study is that it relies on the perception of respondents without the possibility of verifying the source of data. Another limitation of this study is its cross-sectional nature, meaning it cannot be used to establish causative relationships. The quality of data also depends heavily on the accurate reporting of information by respondents, which may be subject to memory errors.

## 5. Conclusions

The results of the presented study indicate a relatively high level of interest and overall acceptance of flu vaccinations among surveyed Poles in the 2022–2023 season. However, at the same time, observed vaccination coverage rates significantly deviated from the targets set by the WHO, especially among individuals aged 18–64 with chronic diseases. The primary concern of the unvaccinated individuals was the perceived lack of effectiveness of flu vaccines. Individuals above the age of 65 were considerably more willing to undergo flu vaccination; however, the actual practice did not fully align with the declarations in this group of respondents. Despite the majority of surveyed respondents being aware of the recommendations for flu vaccinations, especially for those at high risk of flu complications, most respondents under 65 with chronic diseases and nearly half of those above 65 did not consider themselves to meet the criteria for high risk. Therefore, it is suggested that we should strive for further development of the competencies of healthcare professionals and their skills in vaccine promotion. It is the responsibility of the authorities to create optimal working conditions for healthcare workers by providing them with appropriate tools and incentives for their use. An example could be referrals for vaccinations issued by doctors and nurses to individuals at high risk of health complications. Essential to vaccine promotion is communication, which has two dimensions: promotional and organizational. Promoting vaccinations requires coherent and consistent communication about the values and needs of vaccinations, tailored to changing circumstances. This involves planning communication activities for various stakeholders and coordinating them. Communication about influenza and COVID-19 vaccinations should, with varying intensity, continue throughout the year. In the case of the most widespread vaccination, the focus should be on highlighting the benefits of getting vaccinated. The organizational part of communication complements the promotional aspect with practical information on how and where to get vaccinated.

## Figures and Tables

**Figure 1 vaccines-12-00341-f001:**
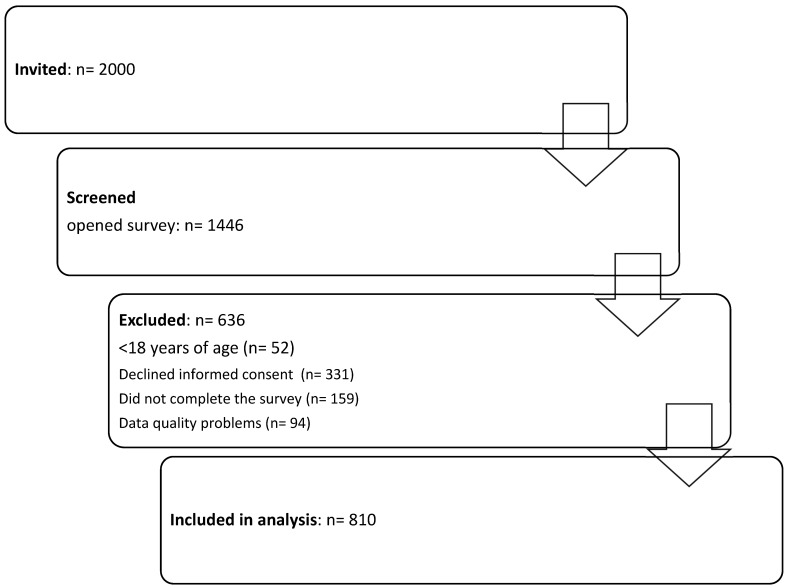
Disposition of survey participants.

**Figure 2 vaccines-12-00341-f002:**
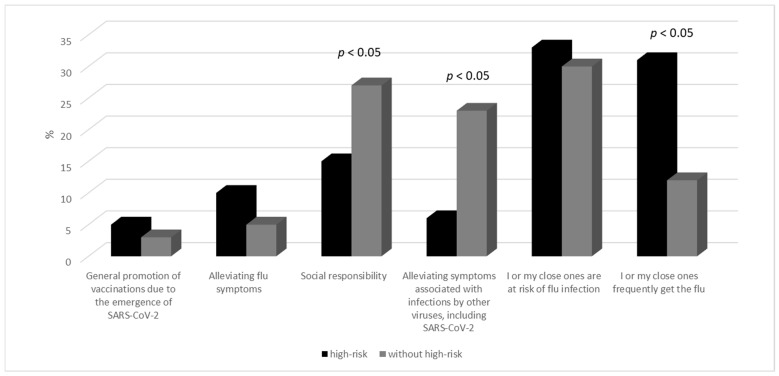
Reasons for getting vaccinated against the flu virus.

**Figure 3 vaccines-12-00341-f003:**
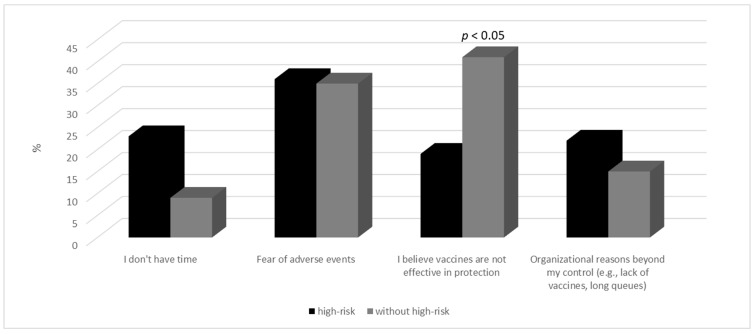
Reasons for not getting vaccinated.

**Table 1 vaccines-12-00341-t001:** Demographic characteristics of respondents (n = 810).

Parameter				General
Age [years; mean (±SD)]				38.32 (3.86)
Sex [female; N (%)]				467 (57.67)
Place of residence			Village	90 (11.12)
			A town with up to 50,000 residents	97 (12.01)
			A town with up to 100,000 residents	197 (24.33)
			A town with up to 250,000 residents	206 (25,42)
			A city with over 250,000 residents	220 (27.12)
Education			Elementary	140 (17.39)
			Vocational school	211 (26.06)
			High school	196 (24.04)
			College/University	263 (32.51)
High-risk group	Without; n (%)	500 (61.77)		
	With; n (%)	310 (38.23)		
			Anemia, thalassemia, hemoglobinopathy	16 (2.05)
			Asthma	60 (7.50)
			BMI > 40 kg/m^2^	28 (3.47)
			Cancer	8 (1.49)
			Chronic CSF leak	5 (0.67)
			Chronic lung disease ^1^	82 (10.12)
			Chronic kidney disease	17 (2.20)
			Chronic liver disease	10 (1.30)
			Hypertension	285 (35.21)
			Heart disease ^2^	100 (12.33)
			Respiratory secretion impairment ^3^	10 (1.21)
			Immune disorder or immune suppression ^4^	4 (0.50)
			Diabetes or other metabolic diseases	122 (15.10)
			Spleen problems or removal	9 (1.09)
		Other high-risk group	≥65 years	195 (24.10)
			Pregnant	15 (1.90)
			Long-term care resident	8 (1.00)
			Indigenous ancestry	15 (1.87)

^1^ Including emphysema, chronic bronchitis, or cystic fibrosis; ^2^ including coronary heart disease, heart failure, and heart attack; ^3^ including increased risk of aspiration; ^4^ including chemotherapy, radiation, steroid use, or an organ transplant. Abbreviations: BMI—body mass index; CSF—cerebrospinal fluid.

**Table 2 vaccines-12-00341-t002:** Assessment of knowledge and practice towards influenza vaccination (n = 810).

	Response [N (%)]		High-Risk (%)		Age (%)				Education (%)			
			with	without	≤20	21–40	41–59	≥65	Elementary	Vocational School	High School	College/ University
How do you understand the phrase “40–70% vaccine effectiveness against the flu”? (n = 810)	Correct answer [421 (52.00)]		66.08	33.92	33.23	42.01	18.78	5.98	5.21	13.55	33.12	48.12
	*p*-value		*p* < 0.0001 *		≤20 vs. 21–40: *p* = 0.1116 ≤20 vs. 41–59: *p* = 0.0224 ≤20 vs. 65: *p* = 0.0057 * 21–40 vs. 41–59: *p* = 0.0003 * 21–40 vs. ≥65: *p* = 0.0188 41–59 vs. ≥65: *p* = 0.0005 *				Elementary vs. vocational: *p* = 0.3022 Elementary vs. High school: *p* = 0.0089 Elementary vs. College: *p* = 0.0002 * Vocational vs. High school: *p* = 0.0053 * Vocational vs. College: *p* < 0.0001 * High school vs. College: *p* = 0.0058 *			
Do you know what the risk groups for influenza infection are? (n = 810)	Yes [575 (71.00)]		77.83	22.17	15.16	33.76	34.44	16.64	2.28	9.98	25.15	62.77
	*p*-value		*p* < 0.0001 *		≤20 vs. 21–40: *p* = 0.0013 * ≤20 vs. 41–59 *p* = 0.0009 * ≤20 vs. 65 *p* = 0.7852 21–40 vs. 41–59 *p* = 0.8871 21–40 vs. ≥65 *p* = 0.0024 * 41–59 vs. ≥65 *p* = 0.0016 *				Elementary vs. Vocational: *p* = 0.3703 Elementary vs. High school: *p* = 0.0616 Elementary vs. College: *p* < 0.0001 * Vocational vs. High school: *p* = 0.0170 Vocational vs. College: *p* < 0.0001 * High school vs. College: *p* < 0.0001 *			
Have you been confirmed to have a SARS-CoV-2 infection in the last season (2022–2023)? (n = 810)	Yes [543 (67.00)]		67.09	32.91	2.69	25.12	28.10	44.09	26.49	23.98	24.03	25.50
	*p*-value		*p* < 0.0001 *		≤20 vs. 21–40 *p* = 0.0577 ≤20 vs. 41–59 *p* = 0.0380 ≤20 vs. 65 *p* = 0.0023 * 21–40 vs. 41–59 *p* = 0.5682 21–40 vs. ≥65 *p* = 0.0003 * 41–59 vs. ≥65 *p* = 0.0015 *				Elementary vs. Vocational: *p* = 0.6337 Elementary vs. High school: *p* = 0.6406 Elementary vs. College: *p* = 0.8500 Vocational vs. High school: *p* = 0.9925 Vocational vs. College: *p* = 0.7733 High school vs. College: *p* = 0.7806			
Have you been confirmed to have a flu virus infection in the last season (2022–2023)? (n = 810)	Yes [291 (36.00)]		68.77	31.23	7.62	17.85	35.43	39.10	7.00	21.77	23.80	47.43
	*p*-value		*p* < 0.0001 *		≤20 vs. 21–40 *p* = 0.2583 ≤20 vs. 41–59 *p* = 0.0101 ≤20 vs. 65 *p* = 0.0043 * 21–40 vs. 41–59 *p* = 0.0245 21–40 vs. ≥65 *p* = 0.0072 * 41–59 vs. ≥65 *p* = 0.5776				Elementary vs. Vocational: *p* = 0.1359 Elementary vs. High school: *p* = 0.0983 Elementary vs. College: *p* = 0.0006 * Vocational vs. High school: *p* = 0.7814 Vocational vs. College: *p* = 0.0006 * High school vs. College: *p* = 0.0010 *			
Have you ever received the flu vaccine? (n = 810)	Yes [421.2 (52.01%)]	Ever in the past [40 (26.79)]	70.01	29.99	6.75	5.07	18.87	69.31	24.12	51.23	13.09	11.56
			*p* = 0.0188		≤20 vs. 21–40 *p* = 0.9386 ≤20 vs. 41–59 *p* = 0.6250 ≤20 vs. 65 *p* = 0.0332 21–40 vs. 41–59 *p* = 0.6370 21–40 vs. ≥65 *p* = 0.0663 41–59 vs. ≥65: *p* = 0.0156				Elementary vs. Vocational: *p* = 0.1723 Elementary vs. High school: *p* = 0.6222 Elementary vs. College: *p* = 0.6030 Vocational vs. High school: *p* = 0.1240 Vocational vs. College: *p* = 0.1451 High school vs. College: *p* = 0.9448			
		During the 2022–2023 flu season [48 (32.12)]	81.08	18.92	12.65	19.95	29.13	58.27	26.09	24.98	24.43	24.50
			*p* = 0.0003 *		≤20 vs. 21–40: *p* = 0.7125 ≤20 vs. 41–59: *p* = 0.4338 ≤20 vs. 65: *p* = 0.0432 21–40 vs. 41–59: *p* = 0.6266 21–40 vs. ≥65: *p* = 0.0464 41–59 vs. ≥65: *p* = 0.0842				Elementary vs. Vocational: *p* = 0.9514 Elementary vs. High school: *p* = 0.9271 Elementary vs. College: *p* = 0.9302 Vocational vs. High school: *p* = 0.9761 Vocational vs. College: *p* = 0.9792 High school vs. College: *p* = 0.9970			
		In the previous 2021–2022 flu season [61 (41.09]	68.22	31.78	4.86	16.07	23.09	55.98	20.08	26.12	27.12	26.68
			*p* = 0.0071 *		≤20 vs. 21–40: *p* = 0.6197 ≤20 vs. 41–59: *p* = 0.4727 ≤20 vs. 65: *p* = 0.0894 21–40 vs. 41–59: *p* = 0.6726 21–40 vs. ≥65: *p* = 0.0262 41–59 vs. ≥65: *p* = 0.0378				Elementary vs. Vocational: *p* = 0.7093 Elementary vs. High school: *p* = 0.6665 Elementary vs. College: *p* = 0.6851 Vocational vs. High school: *p* = 0.9490 Vocational vs. College: *p* = 0.9713 High school vs. College: *p* = 0.9776			
Did you experience any adverse effects from receiving the flu vaccine? (n = 149)	Yes [42 (28.08)]		43.44	55.56	7.62	28.98	19.10	43.30	19.95	21.30	27.32	31.43
	*p*-value		*p* = 0.4343		≤20 vs. 21–40: *p* = 0.4430 ≤20 vs. 41–59: *p* = 0.6434 ≤20 vs. 65: *p* = 0.2390 21–40 vs. 41–59: *p* = 0.6173 21–40 vs. ≥65: *p* = 0.4275 41–59 vs. ≥65: *p* = 0.2350				Elementary vs. Vocational: *p* = 0.9372 Elementary vs. High school: *p* = 0.7112 Elementary vs. College: *p* = 0.5652 Vocational vs. High school: *p* = 0.7558 Vocational vs. College: *p* = 0.6000 High school vs. College: *p* = 0.8260			
How severe were the adverse effects you experienced after receiving the flu vaccine? (n = 42)	Mild [27 (65.28)]		63.89	36.11	26.12	1.77	33.65	38.46	30.72	29.36	18.74	21.18
	*p*-value		*p* = 0.1622		≤20 vs. 21–40: *p* = N.D. ≤20 vs. 41–59: *p* = 0.7512 ≤20 vs. ≥65: *p* = 0.5954 21–40 vs. 41–59: *p* = N.D. 21–40 vs. ≥65: *p* = N.D. 41–59 vs. ≥65: *p* = 0.8275				Elementary vs. Vocational: *p* = 0.9527 Elementary vs. High school: *p* = 0.6324 Elementary vs. College: *p* = 0.4113 Vocational vs. High school: *p* = 0.6682 Vocational vs. College: *p* = 0.7294 High school vs. College: *p* = 0.9199			
	Moderate [13 (31.22)]		62.10	37.90	20.62	21.10	28.17	30.11	17.20	25.99	29.81	27.00
	*p*-value		*p* = 0.3549		≤20 vs. 21–40: *p* = 0.9906 ≤20 vs. 41–59: *p* = 0.8488 ≤20 vs. ≥65: *p* = 0.8049 21–40 vs. 41–59: *p* = 0.8587 21–40 vs. ≥ 65: *p* = 0.8149 41–59 vs. ≥65: *p* = 0.9555				Elementary vs. Vocational: *p* = 0.8176 Elementary vs. High school: *p* = 0.7387 Elementary vs. College: *p* = 0.7989 Vocational vs. High school: *p* = 0.9115 Vocational vs. College: *p* = 0.9776 High school vs. College: *p* = 0.9351			
	Severe [1 (3.50)]		100	0	12.66	12.32	34.12	40.09	28.17	33.10	24.88	13.85
			-		*p* = N.D.				*p* = N.D.			
Did you receive both: the flu and SARS-CoV-2 vaccines during the 2022–2023 season? (n = 810)	Yes [140 (17.28)]		81.08	18.92	16.51	16.69	29.10	37.70	24.10	17.99	26.48	31.43
	*p*-value		*p* < 0.0001 *		≤20 vs. 21–40: *p* = 0.9869 ≤20 vs. 41–59: *p* = 0.2633 ≤20 vs. ≥65: *p* = 0.0678 21–40 vs. 41–59: *p* = 0.2682 21–40 vs. ≥65: *p* = 0.0695 41–59 vs. ≥65: *p* = 0.3879				Elementary vs. Vocational: *p* = 0.5746 Elementary vs. High school: *p* = 0.8193 Elementary vs. College: *p* = 0.4797 Vocational vs. High school: *p* = 0.4362 Vocational vs. College: *p* = 0.2244 High school vs. College: *p* = 0.6254			
How many doses of the SARS-CoV-2 vaccine have you received during the season 2022–2023? (n = 140)	1 [12 (8.53)]		41.01	58.99	26.12	1.77	33.65	38.46	24.12	51.23	13.09	11.56
	2 [13 (9.12)]		33.87	66.13	41.23	12.38	27.15	19.24	29.09	31.65	39.26	36.47
	3 [80 (57.02)]		91.10	8.90	3.69	29.12	33.43	33.76	44.10	38.12	17.78	28.98
	4 [35 (25.33)]		87.28	12.72	12.65	19.95	29.13	58.27	26.49	23.98	24.03	25.50
In your opinion, does the flu vaccine you’ve received alleviate the symptoms of a SARS-CoV-2 infection? (n = 140)	Yes [94 (67.09)]		71.99	28.01	4.86	16.07	23.09	55.98	20.08	26.12	27.12	26.68
	*p*-value		*p* = 0.0001 *		≤20 vs. 21–40: *p* = 0.5625 ≤20 vs. 41–59: *p* = 0.4050 ≤20 vs. ≥65: *p* = 0.0485 21–40 vs. 41–59: *p* = 0.6048 21–40 vs. ≥65: *p* = 0.0064 * 41–59 vs. ≥65: *p* = 0.0108				Elementary vs. Vocational: *p* = 0.6433 Elementary vs. High school: *p* = 0.5948 Elementary vs. College: *p* = 0.6167 Vocational vs. High school: *p* = 0.9369 Vocational vs. College: *p* = 0.9645 High school vs. College: *p* = 0.9720			
Do you plan to get regular flu vaccinations in the future seasons? (n = 810)	Yes [211 (26.12)]		46.79	24.81	12.66	12.32	34.12	40.09	31.76	38.44	29.80	0
	*p*-value		*p* = 0.0087		≤20 vs. 21–40: *p* = 0.9704 ≤20 vs. 41–59: *p* = 0.0376 ≤20 vs. ≥65: *p* = 0.0097 21–40 vs. 41–59: *p* = 0.0345 21–40 vs. ≥65: *p* = 0.0088 41–59 vs. ≥65: *p* = 0.4422				Elementary vs. Vocational: *p* = 0.3977 Elementary vs. High school: *p* = 0.8097 Elementary vs. College: *p* = N.D. Vocational vs. High school: *p* = 0.2821 Vocational vs. College: *p* = N.D. High school vs. College: *p* = N.D.			
	No [373 (46.00)]		22.12	40.09	19.16	33.76	30.44	16.64	33.10	37.32	20.58	9.58
	*p*-value		*p* = 0.0057 *		≤20 vs. 21–40: *p* = 0.0293 ≤20 vs. 41–59: *p* = 0.0898 ≤20 vs. 65: *p* = 0.7058 21–40 vs. 41–59: *p* = 0.5834 21–40 vs. ≥65: *p* = 0.0141 41–59 vs. ≥65: *p* = 0.0453				Elementary vs. Vocational: *p* = 0.4758 Elementary vs. High school: *p* = 0.0568 Vocational vs. College: *p* = 0.0062 * Vocational vs. High school: *p* = 0.0115 High school vs. College: *p* = 0.0016 * Vocational vs. High school: *p* = 0.1528			
	Not sure [226 (27.88)]		31.09	35.10	26.12	1.77	33.65	38.46	40.10	36.12	17.78	6.00
	*p*-value		*p* = 0.6040		≤20 vs. 21–40: *p* = 0.3244 ≤20 vs. 41–59: *p* = 0.1602 ≤20 vs. ≥65: *p* = 0.0441 21–40 vs. 41–59: *p* = 0.1830 21–40 vs. ≥65: *p* = 0.1369 41–59 vs. ≥65: *p* = 0.5239				Elementary vs. Vocational: *p* = 0.5928 Elementary vs. High school: *p* = 0.0126 Elementary vs. College: *p* = 0.0165 Vocational vs. High school: *p* = 0.0385 Vocational vs. College: *p* = 0.0306 High school vs. College: *p* = 0.3000			

N.D.—no data. In the table, the level of statistical significance has been set at *p* = 0.0083 (using the Bonferroni correction) to avoid an excessive number of Type I errors. * Data significant at *p* < 0.0083 (after the Bonferroni correction).

## Data Availability

The datasets used and/or analyzed during the current study are available from the corresponding author upon reasonable request.

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
