# Peer review of "Analysis of Attitudes and Practices towards the Influenza Vaccine in High-Risk Adults in Poland"

_vaccines, 2024, doi:10.3390/vaccines12030341_

Round 1

Reviewer 1 Report

Comments and Suggestions for Authors

Dear authors,

Thank you for the interesting paper. Here are some comments for your consideration:

Abstract:

Change the number of participants from 1446 to the actual number of participants, which is 810.

Introduction:

  1. 1. Lines 50-54 repeat themselves in lines 54-59.
  2. 2. Please include studies that examined the research variables (see, for example, https://doi.org/10.3390/ejihpe14010003).
  3. 3. What were the research hypotheses?
  4. Methods:
  5. In general, the methods section is too brief and needs expansion.
    1. 1. On which social media platforms were the questionnaires distributed? To which email addresses? Who provided the list of addresses? Who disseminated the link?
    2. 2. How did you estimate the survey response rate? Considering that the survey was distributed on social media...
    3. 3. Where were the questions taken from? How were they validated? Please specify the number of questions for each variable and the response scale.
      1. 4. Was the survey anonymous? What explanation was provided to potential participants?
      2. Results:
      3. Please add sub-headings.
      4. Discussion:
      5. 1. There is almost no reference to the literature in the field or a description of findings from previous research. Please add relevant studies.
      6. 2. Add practical recommendations to increase vaccination rates beyond "educational programs." For example, vaccinations at workplaces. Employers will also benefit from this. How can the gap between attitudes and actual behavior be narrowed?

Author Response

Thank you very much for your valuable feedback. We have attempted to incorporate most of them into the manuscript. We believe that taking your suggestions into account will further enhance the substantive value of our work. Attached, please find our references to your comments.

Reviewer 2 Report

Comments and Suggestions for Authors

THIS IS A PAPER ON AN IMORTAT PUBLIC HEATH ISSUE, HOWEVER SOEM CLARIFICATIONS NEED TO BE MADE ESPECIALLY IN THE METHODOLOGY SECTION.

PRGRAPH IN TEXT "The vaccination coverage rate in other developed countries is estimated to be at an 50

average level of around 70%. Reports from Poland suggest that the attitude of Poles to- 51

wards influenza vaccinations does not align with the actual vaccination coverage, as de- 52

spite a positive opinion regarding the effectiveness and importance of vaccinations, the 53

percentage of vaccinated individuals remains low[10]. The vaccination coverage rate in 54

other developed countries is estimated to be at an average level of around 70%. Reports 55

from Poland suggest that the attitude of Poles towards influenza vaccinations does not 56

align with the actual vaccination coverage, as despite a positive opinion regarding the 57

effectiveness and importance of vaccinations, the percentage of vaccinated individuals re- 58

mains low[11]."

PLEASE REPHRASE, THERE IS REPETITION.

The low 70 level of vaccination coverage thus indicates a low level of public awareness of the dangers 71 of not being vaccinated. THIS MAY ALSO MEAN THAT THE PUBLIC DOES NOT COMPREHEND THE VACCINATION BENEFITS, HENCE LACK OF HEALTH EDUCATION AND HEALH PROMOTION? PLEASE COMMENT

PLEASE SPECIFY THE TARGET OPULATION FOR THE SURVEY AND THE STUDY PARTICIPANTS SELECTION PROCESS

"The survey response rate was 56%. Out of 1446 completed questionnaires, 810 re- 99 spondents were included for further analysis. 636 respondents were excluded for further 100 analysis based on exclusion criteria (Fig.1). "

56% IS THE FULLY COMPLETED QUETIONNAIRES. TO CALCULATE THE RESPONSE RATE WE NEED THE TOTAL TARGET GROUP THAT RECEIVED THE QUESTIONNAIRE.

DATA FROM FIGURE 2 STATES 2,000 WERE INVITED IN WHICH CASE THE RESPONSE RATE IS 40.5%. PLEASE ADJUST THE TEXT ACCORDINGLY.

HOW WERE THE 2,000 IDENTIFIED?

PLEASE MAKE A COMMENT ON THE SYNTHEISIS OF THE RESPONDENTS. THE AUTHORS REFER TO THEM AS POLES. ARE THEY ALL POLISH OR DO SOME BELONG TO OTHER ETHNIC GROUPS?

ARE IN THE COUNTRY PUBLIC OUTREACH CAMPAIGNS ON THE FLU VACCINATION BENEFITS ORGANISED? IF SO WHEN?

HOW EXPENSIVE IF THE FLU VACCINE IN POLAND? APART FROM HIGH RISK GROUP ARE THERE ANY SOCIAL CRITERIA FOR FREE FLU VACCINATION? E.G. FOR LOW INCOME OR UNINSURED INDIVIDIALS?

PLEASE MAKE A COMMENT ON THE FINANCIAL HARDSHIP OF PEOPLE IN POLAND CURRENTLY AND THEIR PRIORITIES. ARE THERE ANY DATA FROM CONSUMER STUDIES? 

Comments on the Quality of English Language

SOEM ENGLISH EDITING IS NECESSARY, PLEASE CHECK CLOSELY E.G.

636 respondents were excluded FROMr further 100 analysis based on exclusion criteria

Similar observations were made by authors IN other studies

Author Response

(The authors gave the same response as above.)

Reviewer 3 Report

Comments and Suggestions for Authors

The manuscript is readable and certainly addresses an important topic. Some adjustments are needed to enhane the opportunity for the research to have broader impact.

Perhaps the primary concern is the likely lack of representativeness of the sample relative to the population. It certainly way overrepresents women and does not seem to have the characteristics of a valid random sample.  More information could be provided to argue for representativeness, although a better strategy would be to use post-stratification to reorient the data closer toward representativeness and re-estimate the relevant models.

A related point is that the document reports a very substantial number of p-valuues. These almost certainly overstate the true elvel of statistical significance due to inflated Type I error. False discovery rate adjutment would be ideal, but a Bonferroni correction could help. 

Very little is included regarding implications for policy and practice. Friendly advice is to add perhaps a couple of solid paragraphs addressing initiatives to enhance vaccination rates.

A somewhat sensitive issue in the background is whether the now-former hard-right Polish regime had any role in attenuating vaccination rates for political and ideological reasons--and whether the very recent regime change speculatively might have a different effect on the subject of the study.

Comments on the Quality of English Language

The document is quite understandable, so only minimal standard copediting seems needed.

Author Response

(The authors gave the same response as above.)

Reviewer 4 Report

Comments and Suggestions for Authors

This is is a survey on influenza vaccination in a high risk group in Poland.

INTRODUCTION

In line 45 authors use the acronym WHO but dont explain it.

Line 50 “The vaccination coverage rate in other developed countries is estimated to be at an 50 average level of around 70%. Reports”   Cite a reference.

Line 55, the paper says “Reports   from Poland suggest that the attitude of Poles towards influenza vaccinations does” but there is only one reference.ç

Line 60-67 include references.

MATERIAL AND METHODS.

 Indicate which social media, were used.

Please explain what the authors mean by “Indigenous origin” (line 103)

Please confirm that the questionnaire was distributed only in Poland . What was the language of the questionnaire ¿ (It is assumed that polish, but confirmate it)

Rewrite this paragraph” 107

The data 108 distribution pattern was not normal (unlike the Gaussian function). The analysis of the 109 Test for Proportions determined significant differences between % of group results.”

You should say that you test the data to verify if it has a normal distribution.  And that you do a test for comparing proportion . Please specifiy what test did you use (Chi square, Fisher, etc).

RESULTS

In table 2 the authors made many multiple comparisons. Authors should use Bonferroni Corrections. They can compute a new P, or write a note at the foot of the table indicating that using the Bonferroni correction, the signficance level  was fixed 0.0083.[ (0.05) / 6=0.0083)

The results , section should be rewritten taken into account this consideration.

DISCUSSION

Many studies speaks of COVID-19 vaccine fatige, and its relation with vaccination not only covid19 but also other vaccines. Please take into account this in the disccusion an introduce references related iwth this topic.

Author Response

(The authors gave the same response as above.)

Round 2

Reviewer 1 Report

Comments and Suggestions for Authors

The researchers addressed all the comments,

therefore, I recommend publishing the paper.

Reviewer 3 Report

Comments and Suggestions for Authors

Thank you for the thorough revisions.

Please accept my apology for the substantial number of typos in the initial review. I usually copy and paste comments into Word to check for exactly that problem before uploading the response, but obviously failed to do so on this occasion.